# Videogaming Frequency and Executive Skills in Young Adults

**DOI:** 10.3390/ijerph191912081

**Published:** 2022-09-24

**Authors:** Sara Peracchia, Giulia D’Aurizio, Giuseppe Curcio

**Affiliations:** Department of Biotechnological and Applied Clinical Sciences, University of L’Aquila, 67100 L’Aquila, Italy

**Keywords:** gamers, decision making, attention, inhibition control, switching, executive functioning

## Abstract

Many studies have shown that “action” videogames (VG) training can improve various cognitive aspects (such as attention, enumeration skill, problem solving, vigilance, inhibitory control and decision making). Unfortunately, independently by VG genre, little research has been conducted on the relevance of videogaming frequency to modulate cognitive performance. In the present study, we investigated the differences between two groups of young adults (Experienced Gamers and Casual Gamers, respectively, EGs and CGs) in some attentional and executive abilities. To this end, 19 EGs (age 23.21 ± 1.68 years; gaming frequency 46.42 ± 11.15 h/week) and 19 CGs (age 23.10 ± 2.28 years; gaming frequency 1.31 ± 1.76 h/week) were selected and asked to complete a computer-based and customized version of an executive battery (i.e., Attention Network Task, Game of Dice task, Go/NoGo task and Task Switching). The results showed better basic attentional abilities and alertness level (i.e., as indicated by faster reaction times (RTs) and greater accuracy) in EGs compared to CGs. Moreover, EGs showed a more efficient decision making than CGs, particularly evident in risky decisions. Taken together, such results show that an executive functioning improvement can be observed as a consequence of continuous and constant exposure to VG, independently by the specific genre played. These data can be a useful starting point to develop new and innovative executive training protocols, based and inspired to videogames to be applied in clinical populations suffering, for example, from dysexecutive impairment.

## 1. Introduction

Videogames (VGs) over the past decades have become one of the preferred recreation activities. Realistic graphics, exciting stories and compelling content (not always suitable for the age of the target audience) attract all people, without distinction. In the world, a lot of people own VGs devices, and the number of active videogamers worldwide is on the rise [1]: as an example, in the U.S. during the year 2020, VGs players made up more than 214 million, and 64% of adults and 70% of those under 18 regularly play VGs [2]. In Italy, people who played VGs in 2019 made up 17 million (equal to 39% of the population) aged between 6 and 64 years [3]. It is estimated that, by the end of 2021, the total number of players worldwide will be 2.7 billion [1], an increase also encouraged by the possibility of playing with different devices and probably also facilitated by the specific health and social condition induced by the recent COVID-19 pandemics. These considerations, together with the availability of multiple and differentiated genres present on the market, and the development of new “low cost” devices, show the relevance of VGs as an important slice of the world economy.

Such a great diffusion and evolution of VGs has led researchers, in recent years, to focus on the effects of exposure on health [4]. As a first, some of them focused on the negative effects when exposed to violent VGs, reporting an increase in aggressive behaviour and reduced cognition [5], sleep disturbances [6], incidence of depression and anxiety [7] and addiction [8]. Only in a second phase did the researchers started to investigate the potentially positive impact of VG exposure, focusing particularly on mental health, demonstrating that these positive effects seem to occur at cognitive level. In fact, several studies recently showed improvement as a consequence of videogaming in different skills: visual attention [9,10,11,12], speed [13] and efficiency of visual processing [14], accuracy in the avoidance of irrelevant distractors [15] and multisensory discriminatory tasks [16]. Furthermore, some improvements have also been reported in sensory integration skills [16], mental rotation [17] and visual tracking [18]. Other studies turned their attention to higher-level functions, such as attentional control [15] and change detection strategies [19], confirming a positive effect of VGs.

Many of the benefits associated with videogaming have been explained with an attenuated top-down control, enhanced during the identification and selection of stimuli relevant to the task [14,20]. VG can also modulate the components of event-related potential (ERP); it is thought to reflect the endogenous modulation of the competition driven by the stimulation for selection [21], and it has been associated with changes in activation of the involved frontoparietal network, in executive control and in dynamic allocation of cognitive resources [22,23]. In addition, several studies have shown an improvement in players’ shifting capabilities, i.e., the ability to switch from one task to another [24], and lower switch costs [25,26]. An aspect that links all these studies is the type of VG used in the research projects: such effects would appear upon exposure to VGs of a genre called “action”. Action VGs (AVG; i.e., shooter genres, FPS, strategy, MMORPG, etc.) are characterized by complex strategic schemes and require players to identify and select relevant visual information, to choose the correct motor response in order to execute it, to make decisions in a reduced time and to process specific strategies to reach the previously established objectives. Moreover, AVGs demand continuous monitoring of various objectives and the possibility of moving flexibly from one activity to another in response to changing environmental needs. Conversely, no-Action VG are all the other typologies of VG (i.e., puzzle, brain-teaser, riddle, etc.) that basically do not require particular skills and abilities, except observation and response to environmental stimuli.

Nevertheless, it is important to underline that any type of game (digital or otherwise) can have educational and/or strengthening implications because the game is formative in itself. We think of Jean Piaget, who connected games to the cognitive development of a child, or Lev Vygotsky, who considered play essential for the overall development of a child and at the basis of the proximal development zone, even in adulthood. VGs are simply games, with the only variation being electronic. Their electronic nature makes them highly interactive, and as such, they require more multitasking and problem-solving skills, regardless of gender [27,28,29]: as such, can they affect some cognitive components?

Many studies have aimed to compare cognitive performance in gamers and non-gamers to action and non-action VG using different research protocols and obtaining conflicting results [29,30], while few studies have focused their attention on the effects without taking into account the VG genre. In this regard, we decided to take a step backwards compared to the amount of data present in the literature to understand if exposure to any genres of VG can have effects on some cognitive abilities and how important the amount of playing time to VG. 

The first aim of this study was to compare attentive performance, inhibitory control and switching abilities in experienced gamers (EGs) and casual gamers (CGs). In this regard, we hypothesize the existence of a direct relationship between higher exposure to VG and improvement in the attention performance as these aspects of cognition have been found to be more developed in habitual VG players when compared with non-players [31,32]. As a second aim, we also investigated the ability to choose under risk conditions in the same two samples of VG players. Again, we expect a better decision-making performance in EG than in CG. 

## 2. Materials and Methods

### 2.1. Participants

Through notices published on the university campus and on the main social networks, 257 university students were contacted and interviewed. Each of them completed the videogaming habits questionnaire (see below); on the basis of their self-evaluation and after a telephone interview, 38 male students who met the inclusion criteria to participate in the study, namely playing more than 35 h/week or less than 5 h/week, were selected. The 38 students selected were divided into two “extreme” groups: experienced gamers (EGs), who declared having played all types of VG for a minimum of 40 h per week in the past year, and casual gamers, (CGs) who declared having played all kinds of VG for less than 5 h per week. Each group was composed of 19 participants: EGs had a Mage = 23.21 ± 1.68 years and reported to play Mgame = 46.42 ± 11.15 h, while CGs had a Mage = 23.10 ± 2.28 years and reported a Mgame = 1.31 ± 1.76 h. 

### 2.2. Procedure

The participants were invited to the laboratory in the morning (between 10 and 12 a.m.) to complete a battery of computerized tests to assess the attentive components (sustained attention, alert, orientation and conflict), reaction times and accuracy, shifting capacity and decision-making in risky conditions. All tasks are described below.

The whole procedure took place between October 2018 and July 2019, was conducted in accordance with the Declaration of Helsinki and was approved by the Internal Review Board of the University of L’Aquila (#16/2016).

### 2.3. Tasks

Questionnaire about VG playing habits. The questionnaire aims to collect some information on participants such as age, use and frequency of videogaming per week in the last year (estimated in hours), favourite genre of VGs and phase of the day in which they play more frequently. Such data have been used to select the final participants and to assign them to one of the two groups.

Attention Network Task (ANT) [33]. This task was used to test three attentional networks: alerting, orienting, and executive control. This task asks to indicate the direction of the arrow (left or right) by pressing the right or left mouse button. The test has four cue conditions (no cue, centre, double and orienting) and three flanker conditions (congruent, incongruent and neutral). All combinations are randomly presented in four blocks: a bock of practice of 24 trials and three blocks of 96 trials. The dependent variables analysed included Alert Effect, Percentage of response Accuracy, Orientation Effect, Conflict Effect and Average RT for correct answers.

Game of Dice Task (GDT) [34]. GDT is a computerized task used to assess participants’ aversion/attraction to risky decision. Participants are invited to “predict” the outcome of a dice roll by selecting among different options: options with high-probability but low payoff and options with low probability but high payoff. The aim was to increase the initial capital available (i.e., EUR 1000) along 18 rolls of the dice. Dependent variables analysed included Total GDT Score, Total Number Winnings and Losses, Total Number of Safe and Risky Choices, Winnings in Risky and Safe Conditions, Losses in Risky and Safe Conditions.

Go/NoGo. This computerized task asks the participant to respond (Go) or not (NoGo) to the presentation of a specific stimuli. Subjects had to quickly press a key as soon as they saw the stimuli Big Star, Small Star and Big Cross appear (Go), while they had to refrain from answering when the stimulus small cross (NoGo) appeared. The stimuli appeared randomly for 250 ms, preceded by a fixation point (lasted for 300 ms). The task was organized in two blocks: the first as a training session (12 Go stimuli and 2 NoGo stimuli) and the second block as a test (200 Go stimuli and 40 NoGo stimuli). Analysis were carried out on the dependent variables’ mean RTs to Go, d’ (discriminative capacity) and β (criterium/caution) indices.

Task Switching Test [24]. The visual stimulus pairs consisted of a number (1–9, excluding 5) and a geometric figure (square and rhombus). The presentation of each number was preceded by the appearance of a square or a rhombus indicating the task to be carried out: in the Square task, participants pressed the “A” key when an odd number was presented and the “L” key when an even number was presented. In the Rhombus task, participants pressed the “A” key if the number presented was greater than 5 and the “L” key if the number presented was less than 5. The task is organized with the presentation of a black screen, cue and stimuli. The black screen (350 ms) was followed by the appearance of the cue (square or rhombus), which lasted for 300 ms, always presented in the centre of monitor. Immediately inside the cue, the stimulus (number) appeared, which persisted until the response of the experimental subject. Then, a blank interval of 900 ms followed before a new trial started when the participant made a correct response. When the participant made an incorrect response, a beep sounded. The task consisted of a single block of 1300 stimuli that appeared random. The dependent variables on which the analyses were performed were Median RT for Repetition trials, Median RT for Switch trials, Repetition and Switch Errors, and Switch Cost. 

### 2.4. Statistical Analysis

For each task’ dependent variable, Student’s *t*-test was used to compare the data from EGs and CGs group.

## 3. Results

### 3.1. Attention Network Task

The results showed a significant effect (t36 = 2.173; *p* = 0.036) regarding the Alert Effect variable, indicating a higher alert level of EGs compared to CGs (see Figure 1a). Significant differences emerged also for Percentage of response Accuracy (t36 = 3.139; *p* = 0.003), indicating greater accuracy in EGs compared to CGs (depicted in Figure 1b).

No significant effects have been observed for the remaining dependent variables. 

### 3.2. Game of Dice Task

A significant effect has emerged for Total GDT Score variable (t36 = 3.579; *p* = 0.001), which resulted to be greater in EGs than in CGs (see Figure 2). 

Significant differences emerged for the Total Number Winnings (t36 = 4.217; *p* = 0.000) and Total Number Losses (t36 = −3.813; *p* = 0.001) variables: EGs obtained more winnings and less losses than CGs (depicted in Figure 3a,b). 

Significant differences emerged also for the Total Number of Safe Choices (t36 = 2.535; *p* = 0.016) and Total Number of Risky Choices (t36 = −2.419; *p* = 0.021). In particular, EGs tended to take more safe choices (16.05 ± 1.64) and less risky choices (1.84 ± 1.57) with respect to CGs (13.58 ± 3.92 and 4.21 ± 3.96, respectively). 

Finally, significant differences also emerged for the Losses in Risky conditions (t36 = −3.10; *p* = 0.004), which resulted to be lower in the EG group (0.58 ± 0.76) than in CGs (2.95 ± 3.24). Conversely, the amount of winnings in safe conditions (t36 = 2.653, *p* = 0.012) were greater in EGs (8.78 ± 2.12) than in CGs group (6.79 ± 2.50).

No significant effects have been found on the other dependent variables. 

### 3.3. Go/NoGo

Significant differences among EGs and CGs emerged for mean RT Go variable (t36 = −2.223; *p* = 0.033). In particular, EGs showed a reduction in RT to stimuli Go (249.56 ± 26.30 ms) respect CGs (281.05 ± 55.84 ms, see Figure 4). Additionally, the results of the β index showed a significant effect (t36 = 2.445; *p* = 0.02) showing a slightly greater caution in making decisions for EGs (−1.04 ± 0.35) respect to CGs (−1.31 ± 0.24)

No significant difference emerged for the d’ variables.

### 3.4. Task Switching Test

No significant effects have been observed for any of the considered dependent variables. 

## 4. Discussion

An ever-increasing number of studies highlight positive effects on cognition as a consequence of VG practice. The present study aimed to investigate the effects of a prolonged and continuous exposure to any genre of VG on some cognitive abilities such as attention, executive functioning and decision-making. Four computerized tests were administered to two sub-samples of young male students, namely experienced and casual gamers (EGs and CGs, respectively). The results have shown that the massive exposure to VG typical of EGs determines an improvement in the levels of alertness with an increased accuracy in performance, a faster performance at tasks requiring attention and inhibition control, and a strongly better performance in a task of decision making. No effects have been found on switching abilities.

More specifically, the results from ANT indicated a clear alerting effect of videogaming: EGs were faster and, at the same time, more accurate in a task requiring visual attention and vigilance. This could be explained by the continuous exposure to visual and auditory stimuli, being able to keep their arousal system at a high level of functioning in order to respond as quickly and efficiently as possible.

It has been hypothesized that, behind these improvements, there could also be an increase in cognitive control: in this vein, continuous and prolonged exposure to VG would result in a general and constant activation of alert levels that would support the activation of many systems such as the attentional, mnemonic and motor ones. Playing VGs repeatedly and for long and continuous periods of time would allow continuous training and/or training of these faculties. This hypothesis is consistent with the results obtained at Go/NoGo task: EGs showed a better responsivity (with lower RTs) at of β index, which showed that the CGs are more cautious in decision making than EGs: this would explain the greater number of errors of the latter. These results suggest that while some cognitive skills can be trained with prolonged exposure to any kind of VG [35], others do not produce the same effect, especially in cases where the task requires a high level of attention (in this case, maintaining attention on two tasks), making the performances of EGs and CGs similar [36]. This result probably demonstrates that sustained attention can only be developed through exposure to AVG, as demonstrated by some previous works [13,15].

Nevertheless, EGs tend to show better performance than CGs in some cognitive domains. Attention improves the discrimination of the target stimuli by facilitating the filtering activity of the distractor and by accelerating the rate of processing of visual information [37] (Carrasco and McElree 2001). EGs show a better visual sensitivity [38], which requires a high level of attention needed to form precise representations in the short-term visual memory [39]. Since EGs have a more precise representation of the stimulus, they acquire more information than CGs. Furthermore, they have a lower perceptual threshold, which allows them to start first with the processing of visual stimuli. If we consider what has been said so far, together with the fact that EGs have a faster processing of stimuli [40,41], it is easy to suppose better performance of EGs without the need of faster attentional shifting.

The ability to make decisions in risky conditions was also investigated in the present study. The decision is an act that has to do with outcomes that cannot be predicted with certainty. The choices differ in various aspects, implying risks or having to do with outcomes with which some degree of uncertainty is associated. Taking risk involves the possibility of being able to damage something and/or someone but, at the same time, offers the opportunity to obtain different forms of reward [42]. In the present study, we tried to investigate the behaviour of EGs in conditions of risk. Significant improvements in the EGs have emerged on different aspects of the decision-making process under risky conditions, obtaining higher total scores, a higher number of winnings and a lower number of losses, quantitatively lower losses in risky conditions and higher winnings in safe conditions, a greater number of safe choices and a smaller number of risky choices. These, at the present time, are the first data present in the literature that show a greater ability of the EGs to make better choices in conditions of risk to reach a certain objective.

The only negative effect was observed with respect to task switching. This task has been largely used in the literature to compare people with different videogaming experience and results were very variable. While some authors showed a better performance in videogamers [43], others indicated that such a superiority was dependent by age of active onset of videogaming [44], and others concluded that the relationship between the reduction in switch cost and action game playing was causal [26] or totally absent [45]. We believe that such an ability could be not sensitive to activities such as videogaming because highly cognitively complex and structured. Alternatively, we could explain this lack of effect with sample, e.g., [44] or methodological [26] differences.

On the whole, the present results can be explained by referencing that prolonged, continuous and repeated exposure to any kind of VG often involves the parallel execution of different activities and fast and immediate decision making. Playing VGs represents a sort of daily gym for the analysis of stimuli and decision-making activities. This could lead to an improvement in decision-making capacity, which linked to an increase in attentional skills, involves to the rapid processing of all aspects of the present situation and to identifying the most correct and advantageous choice. Therefore, massive exposure to any genre of VG could potentially lead to an improvement in decision-making capacity.

### Limitations to the Study and Future Directions

As a first limitation, it should be highlighted the relatively small sample size that contribute to reduce both the reliability and generalizability of results: it is undoubtedly appropriate and desirable to conduct further investigations with a greater sample size. Moreover, female subjects have not been tested: it would be interesting to evaluate the abilities of EG women and to compare the data obtained with the performances of male EGs. Linked to these two points, there are some issues related to methodological aspects. One is the issue of the criteria used to differentiate the two groups of “hard” and “soft” players. Since in the literature there is a great variability in the choice of frequency ranges, e.g., [46,47], here, we decided to use a differentiation based on the distribution of our initial contacted sample, with the aim to give the best description of our sample and, at the same time, to better take into account the differences between participants with regard to this activity. Another one was the issue linked to the use of computer-based tests: one could argue that there is the possibility of a potential bias. Nonetheless, in the phase of selection, we also controlled for their practice and all appeared “expert” users of computer, defining themselves as confident in computer use. Moreover, all the cognitive-behavioural task we used are very easy and have simple and basic graphics, which could instead paradoxically disadvantage experienced gamers (used to more challenging and stimulating tasks). Furthermore, a longitudinal approach could be of interest by training naive subjects belonging to different age ranges with different kinds of VG to understand if there is a real training effect, in which age range is higher and which kind of VG have this effect. Finally, another possible concern is related to the experimental design, since, unlike the studies in the literature that investigated the causal relationship between VG and differences in cognitive abilities, our study shows that there is a correlation between prolonged exposure to VG and improvements in some cognitive abilities.

The limitations mentioned so far could have produced errors in the results, which could be addressed in future studies.

The experimental investigations previously conducted have not yet allowed for what cognitive skills, in particular, can be improved following exposure to VG to be highlighted. Identifying the effects is really very complex due to the countless variables that are at stake (personality, gender, background, research protocol, countries, etc.). However, these results can be considered as a further proof that prolonged, repeated and constant exposure to VG of all kinds acts as training for some cognitive skills, probably because the VG (of any genre), being an interactive game, provides the necessary stimuli to train cognitive skills. It is probable that “action” VGs train some cognitive faculties such as sustained attention [15] but even simple repeated exposure to any kind of VG can have effects on cognition.

Because of the cross-cutting nature of the present study, it is not possible to determine whether the differences between EGs and CGs are causally related to exposure to VG or are simple reflections of pre-existing group disparity. Under the theoretical point of view, we can definitely put forward the assumption that videogaming and cognitive performance are linked: it remains to be understood if the more cognitively able individuals tend to be involved in videogaming or if such activity train and help to develop mental skills. A model of these possible interactions needs to be worked out. The first hypothesis, that people with superior attentional abilities achieve better results in VG and consequently tend to play more and more often [48,49], has not been confirmed by a growing number of recent evidence showing a causal relationship between frequency of videogaming and performance improvement [23,50,51,52,53,54,55]. Thus, the second hypothesis would be accepted: i.e., that VGs can train and improve cognitive abilities. As a consequence of this, practical significance and applications would significantly change. In this vein, a lot of applications could be developed, from neuropsychological rehabilitation to educational applications, from learning (e.g., driving and working) to cognitive enhancement.

Taken together, these results question the data in the literature, leaving many questions open and encouraging future work.

## 5. Conclusions

An executive functioning improvement can be observed as a consequence of continuous and constant exposure to VG, independent of the specific genre played. The mechanism of action could rely on a sort of “gym” during continuous previous videogaming that contributes to train the neurocognitive abilities of more expert gamers. Present results can be a useful starting point to develop new and innovative executive training protocols, based on and inspired by videogames dedicated to clinical populations with dysexecutive impairment.

## Figures and Tables

**Figure 1 ijerph-19-12081-f001:**
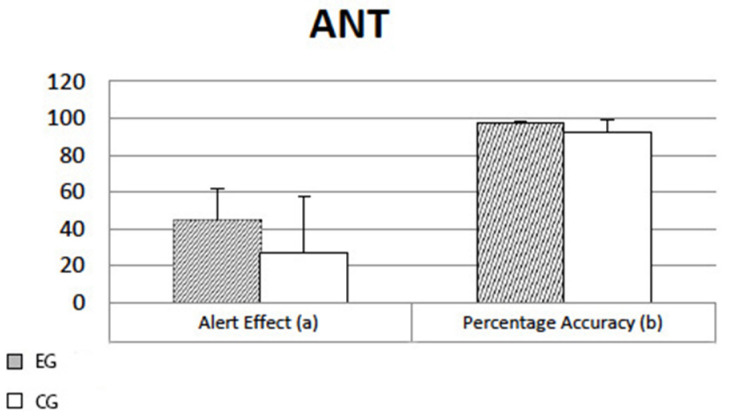
Results at the Attention Network Task (ANT): effects on Alert (panel **a**) and Accuracy Percentage (panel **b**).

**Figure 2 ijerph-19-12081-f002:**
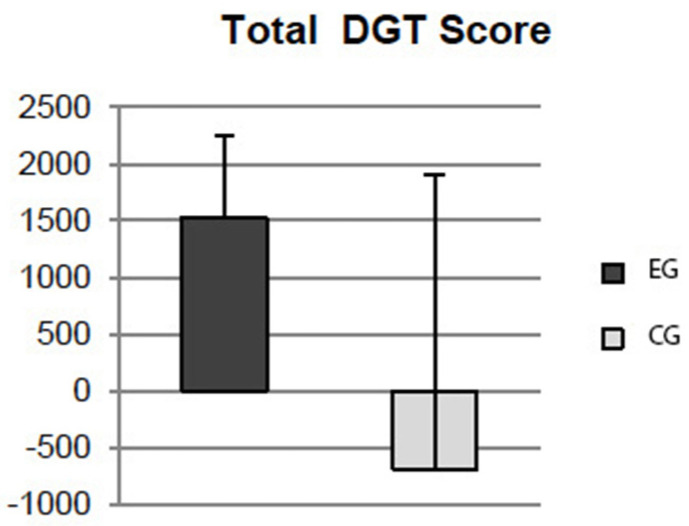
Results at the Game of Dice Task (GDT): effects on Total Score.

**Figure 3 ijerph-19-12081-f003:**
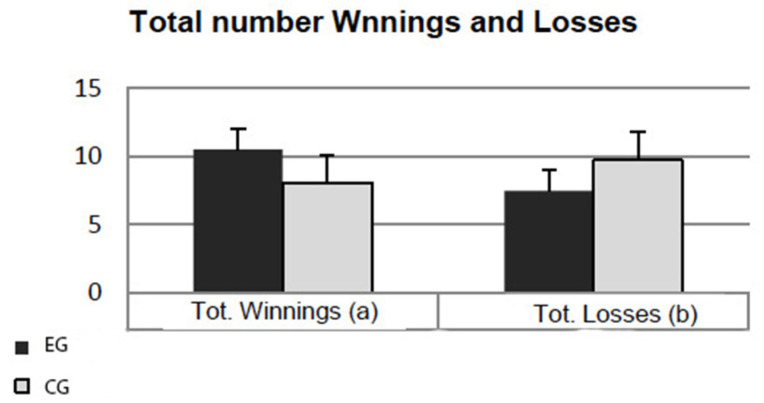
Results at the Game of Dice Task (GDT): effects on Total Number Winnings (panel **a**) and Total Number Losses (panel **b**).

**Figure 4 ijerph-19-12081-f004:**
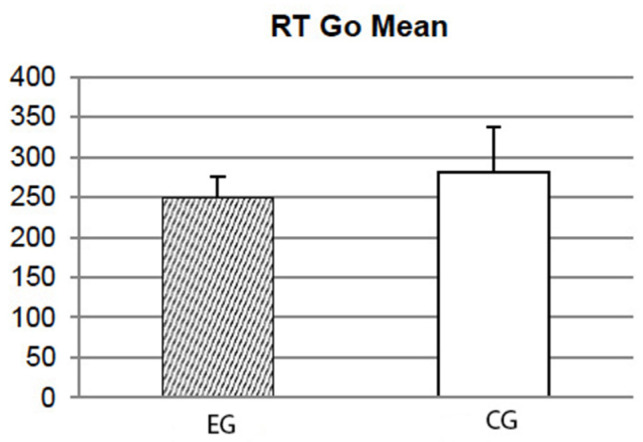
Results at the Go/NoGo task: effects on mean RT to Go stimuli.

## Data Availability

Data included in the article; further inquiries can be directed to the corresponding author.

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
