# Peer review of "Videogaming Frequency and Executive Skills in Young Adults"

_ijerph, 2022, doi:10.3390/ijerph191912081_

Round 1
Reviewer 1 Report
The article "Videogaming Frequency And Executive Skills In Young Adult" has high level of novelty and interesting for readers. But at the same time, while working on the article, I was visited by several questions regarding some aspects of this article. First of all, I had a question why such time periods as less than 5 hours or more than 35 hours were chosen as inclusion criteria? Another question is whether it was correct to use computer-based tests. This created an unequal testing field for experienced and casual players, as experienced players feel more confident when working/playing on a computer. And one more question, is it possible that those who spend more time playing computer games, and initially have higher decision-making abilities, switch from one activity to another and so on. I would be happy to see explanations on these issues in the introduction section and other relevant parts. Thank you!
Author Response
Reviewer 1
The article "Videogaming Frequency And Executive Skills In Young Adult" has high level of novelty and interesting for readers.
Authors’ reply (AR): We thanks the Reviewer for his/her appreciation of the manuscript.
But at the same time, while working on the article, I was visited by several questions regarding some aspects of this article. First of all, I had a question why such time periods as less than 5 hours or more than 35 hours were chosen as inclusion criteria?
AR: Looking at the literature, different authors used very different range to compare “hard” and “soft” players. Some of them considered as hard players those engaged in videogaming around 8-10 hs per week (e.g. Donohue et al, Atten Percept Psychophys (2012) 74:803–809; Wu et al, Front Psychol. 2021 Jun 2;12:611778). Others (such as Latham et al (2013) Front. Psychol. 4:941. doi: 10.3389/fpsyg.2013.0094) considered as experienced videogamers those who “…have played for 20+ h per week over the past 10 years…”
Since this very variable situation, we decided to use a differentiation based on the distribution of our initial contacted sample. Between the 257 students interviewed, we recorded very extreme data, with those declaring small interest for videogaming (Casual, with an use under 5 hs per week) and those with a very pervasive interest toward this activity (Experienced, with more than 35 hs per week).
This seemed the best description of our sample, to better take into account the differences between people with respect to this activity.
Another question is whether it was correct to use computer-based tests. This created an unequal testing field for experienced and casual players, as experienced players feel more confident when working/playing on a computer.
AR: This is a very interesting question and we also discussed about the possibility of a potential bias in the testing phase. Nonetheless this, in the phase of selection we also controlled for the “daily” use of computer in activities like studying, working and so on. Excluding the videogaming activity, all participants were “expert/experienced” users of computer and at the interview defined themselves as confident in computer use.
Moreover, all the cognitive-behavioral task we used are very easy and with a simple and quite “monotonous” graphic, that could instead paradoxically disadvantage experienced gamers!
And one more question, is it possible that those who spend more time playing computer games, and initially have higher decision-making abilities, switch from one activity to another and so on.
AR: The Reviewer’ observation is right, but based on present data, we do not feel we can answer or speculate about a possible switch or not…
I would be happy to see explanations on these issues in the introduction section and other relevant parts.
AR: We included several sentences on Discussion section, to better discuss these aspects, as also requested by Reviewer #1.
Reviewer 2 Report
This paper is based on existing research, aiming to explore the effect of exposure to any genres of VG and the time of playing games on some cognitive abilities. Although the description of the research process and result analysis is concise, the research methods and the provided reference materials are applicable. Compared with a little, the article has relatively large defects.
Pay attention to the use of abbreviations. The abbreviations should be complete for the first time. In the Abstract, in the current 13th line, EGs are omitted; the RT in line 18 is not explained.
It is mentioned in the abstract and introduction that the existing research rarely focuses on verifying its modular cognitive performance through the VG genres. So it is proposed to make up for this research gap. The paper also introduces action VG and non-action VG. However, the overall process of the experiment only requires the participants to have played various types of VG and doesn’t make more profound research on VG types, which is slightly insufficient.
The details of the survey process are not detailed. What is the date of the questionnaire, whether the survey objects are specific (e.g. whether 257 college students are randomly sampled, and what criteria are used to determine 38 students)? As “a prolonged and continuous exposure” mentioned in the discussion, what is the duration of playing any genre of VG in this survey sample? In addition, the survey sample is 19 men in each group, and it universally represents the conclusions. To an extent, it makes people doubtful.
In the discussion, the authors briefly explained the results, and the explanation was very concise. Besides, there was no connection and consideration with existing studies. The results of the task switching test show that the switching capabilities have not been significantly affected, and this paper should explain it accordingly with the actual connection. The whole article does not summarize the theoretical significance, and highlights and innovations of the research are not prominent, which makes people stay in some confirmatory results. Meanwhile, the research data and results are OK, but it does not explain their practical significance and application value. How can the results be applied to reality to improve or enhance the situation? In addition, the citation of references is not very ideal. The number is second. Most of the cited literature is from old previous studies, which can lay a good foundation for the paper. However, the research did not well investigate the recent academic progress. On the other hand, it also verified the novelty and innovation of the study.
On the whole, the research topic is not presented comprehensively. The article’s overall layout is incomplete, and many statements and paragraphs in the paper need a better idea connection.
Author Response
Reviewer 2
Pay attention to the use of abbreviations. The abbreviations should be complete for the first time. In the Abstract, in the current 13th line, EGs are omitted; the RT in line 18 is not explained.
AR: We thank the Reviewer for these notes: the text has been changed accordingly.
It is mentioned in the abstract and introduction that the existing research rarely focuses on verifying its modular cognitive performance through the VG genres. So it is proposed to make up for this research gap. The paper also introduces action VG and non-action VG. However, the overall process of the experiment only requires the participants to have played various types of VG and doesn’t make more profound research on VG types, which is slightly insufficient.
AR: The present study was aimed at investigating the effect of videogaming independently by genre (explicitly stated on lines 88-91).
To avoid a reasonable misunderstanding, we deleted lines 87-88 that probably were the origin of this mistake.
The details of the survey process are not detailed. What is the date of the questionnaire, whether the survey objects are specific (e.g. whether 257 college students are randomly sampled, and what criteria are used to determine 38 students)? As “a prolonged and continuous exposure” mentioned in the discussion, what is the duration of playing any genre of VG in this survey sample? In addition, the survey sample is 19 men in each group, and it universally represents the conclusions. To an extent, it makes people doubtful.
AR: The questionnaire and the study took place between October 2018 and July 2019.
The 257 students were not randomly sampled but were all the respondent we reached “…Through notices published on the university campus and on the main social networks, 257 university students have been contacted and interviewed….” (info already stated on lines 108-109).
The criteria used to select the 38 final sample were th first point raised also by Reviewer #1. As said before, looking at the literature, different authors used very different range to compare “hard” and “soft” players. Some of them considered as hard players those engaged in videogaming around 8-10 hs per week (e.g. Wu et al, Front Psychol. 2021 Jun 2;12:611778). Others (such as Latham et al (2013) Front. Psychol. 4:941. doi: 10.3389/fpsyg.2013.0094) considered as experienced videogamers those who “…have played for 20+ h per week over the past 10 years…” Since this very variable situation, we decided to use a differentiation based on the distribution of our “initial” contacted sample. Between the 257 students interviewed, we recorded very extreme data, with those declaring small interest for videogaming (Casual, with an use under 5 hs per week) and those with a very pervasive interest toward this activity (Experienced, with more than 35 hs per week). This seemed the best description of our sample, to better take into account the differences between people with respect to this activity.
The duration of playing any genre of VG in the survey (not reported in Discussion but in Methods section, line 114) was referred to the last/past year.
Finally, in the limitations the issue of a sample limited to males has been already highlighted (lines 297-299).
All missing info requested in this point by the Reviewer #2 (in some cases coincident with those of Reviewer #1) have been included in the revised manuscript.
In the discussion, the authors briefly explained the results, and the explanation was very concise. Besides, there was no connection and consideration with existing studies. The results of the task switching test show that the switching capabilities have not been significantly affected, and this paper should explain it accordingly with the actual connection.
AR: The revised Discussion includes more explanation of results and their consideration with existing studies. Also, a reflection on task switching have been included. As a consequence, some new references have been included and this could be useful also to answer to some concerns raised by Reviewer #2.
The whole article does not summarize the theoretical significance, and highlights and innovations of the research are not prominent, which makes people stay in some confirmatory results.
Meanwhile, the research data and results are OK, but it does not explain their practical significance and application value. How can the results be applied to reality to improve or enhance the situation?
AR: Also in this case, we tried to include in the revised version of Discussion some sentences to answer to these suggestions.
In addition, the citation of references is not very ideal. The number is second. Most of the cited literature is from old previous studies, which can lay a good foundation for the paper. However, the research did not well investigate the recent academic progress. On the other hand, it also verified the novelty and innovation of the study.
AR: We tried to cite seminal studies, now well accepted by the community. Most recent studies are surely interesting but usually they do not add much more to the current knowledge. Nonetheless, as said in a previous point we included some more (and recent) publications to discuss results on task switching.
We hope these changes go in the direction indicated by the Reviewer; if not, we would accept more punctual suggestions from he/she about recent studies to include in the reference list, aimed at enriching the description of academic progress.
On the whole, the research topic is not presented comprehensively. The article’s overall layout is incomplete, and many statements and paragraphs in the paper need a better idea connection.
AR: The overall layout we applied exactly followed the journal template requests, and we tried to fill all needed sections and paragraphs. Also in this regard, we are happy to receive suggestions from the Reviewer to better organize the paper.
Reviewer 3 Report
Reviewer Report
Manuscript ID: ijerph-1877924
Type of manuscript: Article
Title: Videogaming Frequency And Executive Skills In Young Adults
This study investigates the relevance of of videogaming frequency to modulate cognitive performance. The topic is very interesting and new.
MAIN CONCERN:
The structure of the paper is clear and there is adequate literature and related work documented.
However, as the authors point out, the study has limitations: the sample is very small, which may detract from the reliability of the results.It would be advisable to enlarge the sample in subsequent studies.
Nevertheless, as an initial approach to the topic, the paper is very interesting and the methodology is very appropriate.
MINOR COMMENTS
In the following sentence one word is missing:
In the present study we investigated the differences between two groups of young adults (Experienced and Casual Gamers, respectively and CGs) in some attentional and executive abilities.
Add the acronym EGs (Line 13)
In the present study we investigated the differences between two groups of young adults (Experienced and Casual Gamers, respectively EGs and CGs) in some attentional and executive abilities.
Author Response
Reviewer 3
The topic is very interesting and new.
AR: We thanks the Reviewer for his/her general appreciation of the manuscript.
MAIN CONCERN:The structure of the paper is clear and there is adequate literature and related work documented. However, as the authors point out, the study has limitations: the sample is very small, which may detract from the reliability of the results.It would be advisable to enlarge the sample in subsequent studies.
Nevertheless, as an initial approach to the topic, the paper is very interesting and the methodology is very appropriate.
AR: We strongly thank the Reviewer for his/her positive evaluation of the initial intent of this work and of the appropriateness of methodology.
As stated in the Discussion one of the strongest limitations of the present study is the limited sample size: it is our intention to significantly enlarge the sample in subsequent studies, possibly involving other groups at both national and international level. In this way we could also control for the very variable range of gaming frequency (see the first point raised by Reviewer #1).
As he/she rightly stated, this should be seen as a first approach to the topic.
MINOR COMMENTS
In the following sentence one word is missing:
In the present study we investigated the differences between two groups of young adults (Experienced and Casual Gamers, respectively and CGs) in some attentional and executive abilities. Add the acronym EGs (Line 13)
AR: We thank the Reviewer: the missed acronym was included in the revised version of the manuscript.
Round 2
Reviewer 2 Report
The authors answered the proposed comments one by one. They also adjusted typing errors. But the overall modification didn't meet my requirements. First, the journal template requests are general but can be adjusted appropriately under your paper writing. For example, in the Discussion, the authors explained much but focused on different aspects. Such as research limitations and future research can be used as a new section or sub-section. Second, the study design was only a single complement to the time. Finally, the innovative achievements mentioned are relatively scattered, and the Discussion on the dual research significance of theory and practice needs to be improved. In general, the article has a particular research value, but the novelty is insufficient, and the advantages do not make up for flaws enough, so my suggestion is rejection.
Author Response
Reviewer 2
The authors answered the proposed comments one by one. They also adjusted typing errors.
Authors’ Reply (AR): We thank the Reviewer for his/her appreciation.
But the overall modification didn't meet my requirements. First, the journal template requests are general but can be adjusted appropriately under your paper writing. For example, in the Discussion, the authors explained much but focused on different aspects. Such as research limitations and future research can be used as a new section or sub-section.
Authors’ Reply (AR): To answer to Reviewer’ request, in this revised version we included a new sub-section related to limitations to the study and future directions. The manuscript has been changed accordingly.
Second, the study design was only a single complement to the time.
Authors’ Reply (AR): To answer to Reviewer’ request, in the revised version of the manuscript, we dedicated a sub-section to procedure/study design.
Finally, the innovative achievements mentioned are relatively scattered, and the Discussion on the dual research significance of theory and practice needs to be improved.
Authors’ Reply (AR): We discussed all the results observed in the study in a systematic way, thus we do not understand what the Reviewer means with “scattered”. In the same vein, a whole para has been dedicated to the significance of theory and practice of these results, where we discussed these issues on the light of present data. We deliberately decided to avoid speculation no directly based or linked to our data.
In general, the article has a particular research value, but the novelty is insufficient, and the advantages do not make up for flaws enough, so my suggestion is rejection.
Authors’ Reply (AR): We fully respect the Reviewer idea, with which (of course) we do not agree. From our side, this is and remains a personal point of view, clearly conflicting also with the point of view of the other 2 Reviewers that assessed the significance and value of this contribution and that concluded with an endorsement.
In the present and previous step of reviewing, we answered to all queries of the Reviewer 2, also changing the structure of the manuscript and this was considered not enough.
We think that sentences such as “…the novelty is insufficient, and the advantages do not make up for flaws enough…” or “…But the overall modification didn't meet my requirements” indicate a prejudice toward present study that undermine the reliability of reviewing process.